# The Efforts of Government-Driven Reform of Both State and Personal Rites in Early Chosŏn: A Historical Shift from Spiritual Efficacy to Meritorious Practice

Ch'oe Chongsŏk

Department of Korean History, Dongduk Women's University, Seoul 02748, Republic of Korea; sunnyuni3579@gmail.com

**Abstract:** In the fifteenth century, the government of Chosŏn, Korea, influenced by the new religious understandings from early Ming China, strived to reform state and personal rites by eliminating elements of spiritual efficacy to align with contemporary religious perspectives. This series of ritual reforms mirrored various Ming policies that, despite being appealing theoretically, struggled with implementation due to a mismatch with local and personal realities. This suggests that Chosŏn's government-led reforms, diverging from traditional beliefs, faced similar challenges in Korea, leading to various problems. This study delves into the ritual transformations prompted by the reforms in the early Chosŏn era. It highlights the government's partial success in reforming state rites by removing elements of spiritual efficacy despite potential hindrances and deviations from traditional practices. However, this study also notes the failure of reforms concerning personal rites, which did not yield significant results. It reflects on the complexities and implications of these reforms, considering the socio-religious context of the time and the influence of contemporary Ming China.

**Keywords:** Chosŏn government ritual reforms; Ming China religious influence; state rites; personal rites; elements of spiritual efficacy

## 1. Introduction

This study investigates the efforts of the early Chosŏn 朝鮮 government (1392–1910) to reform both state and personal rites,[1] specifically aiming to exclude element of spiritual efficacy (*yŏnghŏm* 靈驗)[2] from these rites in favor of a newly accepted concept of religion. Specifically, this analysis contends that the early Chosŏn government's reform efforts, driven by a shift towards Neo-Confucianism and influenced by contemporary Ming policies, were met with limited success. This was due to the intrinsic challenges of aligning these reforms with the prevailing socio-religious fabric of Chosŏn society, thereby leading to a nuanced outcome where some objectives were met, while others were not, reflecting the complex interplay between ideology, tradition, and societal realities.

During the Koryŏ 高麗 period (918–1392), gods were mostly anthropomorphic beings that possessed spiritually potent power, and rituals to these gods were acts of soliciting what sponsors wished by relying on the gods' miraculously effective power. In contrast, the gods in Chosŏn, especially those worshipped in public rites, were beings that granted benefits to the state and its people.[3] And rituals also came to refer to acts that should be performed as a reward to the gods for their bestowed benefit. In the early Chosŏn period, there was demand from Neo-Confucian scholar-officials to reform the state rites in order to achieve new ideological orientations and eliminate the remnants of Koryŏ's rituals, such as removing the rites that relied on spiritual efficacy from the state rite system. In addition, with the onset of the Chosŏn era, the government emphasized the importance of conducting rites appropriate to the status of individuals (*myŏngbun* 名分), and enforced rules where officials and the general populace no longer performed rites to the gods of mountains and

rivers, because only a king was allowed to conduct such ceremonies. Instead, they were to only conduct ancestral rites according to their own status.

However, it was not easy for the Chosŏn government to enforce these reforms. As will be shown, the early Chosŏn government took this action by accepting the new religious perception that emerged in the Ming dynasty, based on Neo-Confucianism,[4] as well as various measures accompanying it. These policies, however, were not enforced easily, even in Ming, as they did not consider the reality in local societies. Based on this fact, it can be said that the government-driven reform in early Chosŏn of state and personal rites, based on a new religious perception that was different from the traditional one, was too idealistic to enforce and take root in the Korean society of the time, causing various problems.

The government-led reform of early Chosŏn in the realm of ritual practices has been understood solely from the perspective of Confucianization. According to this perspective, the wholesale project of Confucianization was conducted on the foundation of Chosŏn and, as a part of this project, various reform policies, such as expanding and strengthening Confucian rites and abolishing non-Confucian ones (Han 1976; Deuchler 1992; Han 2002; Kim 2003; Kim 2007; Yi 2009). According to the traditional perspective that Confucianization took place in the early Chosŏn period, Buddhist rituals were removed from the state rite system, and the Buddhist faith of officials and the general populace was suppressed.[5] However, these traditional perspectives or views represent an overgeneralization of the situation and lack concrete context. As will be shown, studies based on the Confucianization perspective hardly pay attention to the contexts in which the government attempted to enforce top-down reform, with its new understanding of religion, in situations where people still relied on the gods' *spiritual efficacy* to fulfill their wishes. Furthermore, these studies often overlook the problems caused by these discrepancies.

Considering these contexts and problems, this paper examines the features and consequences of the reform of state and personal rites in early Chosŏn. In particular, this paper focuses on how a reform that went beyond traditional practice was possible despite the various issues that could have caused it to fail, and how the reform eventually came to fruition in a sense that the government successfully removed the element of spiritual efficacy from the state rites (Han 2002; Kim 2003). This paper also explores why the government's reforms of personal rites failed, despite its efforts, without yielding significant outcomes.

## 2. Koryŏ Period: The Operation of the Rites Based on Spiritual Efficacy

I will first examine the religious practices of the Koryŏ dynasty that became the targets of early Chosŏn's reforms. The Koryŏ state rite system was largely based on the Tang Chinese system (Yi 1991). It consisted of three different levels: the great rite (*taesa* 大祀), medium rite (*chungsa* 中祀), and small rite (*sosa* 小祀), mirroring the structure of the Tang system. The rites within these three categories were similar to those in Tang. Nonetheless, unlike the Tang system, there were quite a number of rites in the Koryŏ system that did not belong to these three categories.[6] It is unclear how these uncategorized rites were referred to during Koryŏ, though during Chosŏn, they were derogatorily designated as miscellaneous rites (*chapsa* 雜祀).[7] While this term was not employed in Koryŏ, it is used in this paper specifically to refer to the state rites not categorized as great, medium, or small rites, for the convenience of discussion.[8]

Mountain–river rites, which occupied a large portion of the *chapsa* during Koryŏ, demonstrate how the state rite system (*sajŏn* 祀典, lit. rite register) revolved around the notion of spiritual efficacy at the time. Among various mountain–river rituals, certain rites were designated state rites, thereby gaining the status of national mountain–river ceremonies, because they had something special, showing a spiritual efficacy that was recognized by the government (Kang 2005, pp. 157–65; Pak 2012).

The biography of Ham Yuil 咸有一 in the *Koryŏsa* 高麗史 is one of the rare documents that detail how a mountain–river rite could be selected as a state rite.[9] It also highlights the importance of the spiritually potent power of gods among the people, and how the Koryŏ



government recognized such gods, registering some into the *sajŏn* based on their power. The biography describes Ham, a central government official, as having "torn down the shrines on several mountains that did not display spiritual efficacy." This action indirectly reflects the contemporary sentiment that shrines displaying spiritual efficacy were considered significant and, thus, were preserved.

In the context of Koryŏ, where the spiritual efficacy of the gods was highly valued, the central government frequently tested gods believed to be miraculously effective in local societies to determine whether they indeed possessed such qualities and, if so, the extent of their spiritual efficacy. Ham's biography reports many instances where he tested the spiritually potent power of locally revered gods. I present a notable case relevant to the registration in the *sajŏn*:

> Ham was appointed as a magistrate in *Hwangju* 黃州. Within his jurisdiction in *Pongju* 鳳州, there was a pond known as spiritually potent, located beneath Owl Rock (*Bu-ung-i pawi*). Ham gathered villagers and filled the pond with trash. Suddenly, clouds gathered, and heavy rain, accompanied by thunder and lightning, poured down, frightening the people. Shortly afterward, the sky cleared, and all the trash from the pond was swept to the top of a distant hill. Upon hearing of this event, the king sent a close retainer to perform a rite and had the pond registered in the *sajŏn*.

Local magistrate Ham Yuil, in his jurisdiction, tested the spiritually potent power of the god associated with a pond under Owl Rock by filling the pond with trash. Impressively, the deity demonstrated its power and passed the test. Upon receiving the report, the king dispatched an official to perform a rite and officially register the deity into the *sajŏn*. It is likely that the local community, for a long time, has relied on the miraculous power of the pond (deity) under Owl Rock to fulfill their wishes. Upon confirming that this pond (deity) possessed the ability to exhibit impressive spiritual efficacy, the state also sought to depend on this entity when necessary, leading to its registration in the *sajŏn*.

Considering that not all mountain–river gods were selected for the *sajŏn*, even after passing spiritual efficacy-tests administered by central officials, it appears that simply passing this test was not the sole criterion for eligibility in the *sajŏn* register. More likely, gods that demonstrated rather striking spiritual efficacy in the tests were chosen. Another case involving Ham supports this view: In *Tŭngju*, the local guardian deity (*sŏnghwang* 城隍) repeatedly possessed shamans and accurately foretold the nation's fortunes and misfortunes. Ham then conducted a state rite at this deity's shrine, suggesting that gods displaying exceptional spiritual efficacy were indeed registered and became the subjects of state rites.[10]

Based on the operation of the *chapsa* rites, it can be inferred that spiritual efficacy or spiritually potent power was a primary factor in the functioning of the *sajŏn* in Koryŏ. This approach persisted into the early Chosŏn dynasty with little change (Ch'oe 2009b, pp. 212–30). Such operation of the *chapsa* likely reflected the social milieu of the time, where individuals or communities publicly performed acts, most notably rites, to fulfill their desires by relying on gods they believed to be miraculously powerful. In this context, the government tested whether these gods truly possessed spiritually effective power and the extent of that power. Presumably, the government sometimes removed the shrines of gods whose power could not be verified, while at other times, it registered those with notably strong powers into the state rite system.

## 3. The Reign Periods of King T'aejo and King Chŏngjong: The Attempt to Reform State and Personal Rites and Its Failure

Less than a month after the founding of the Chosŏn dynasty, Cho Pak 趙璞, the Minister of Rites, and other officials submitted a petition for a comprehensive reform of the state rite system. This was particularly aimed at the *chapsa* rites that had not been categorized as great, middle, or small rites and had mostly been selected for their spiritual efficacy.[11] This petition was a result of Chosŏn officials embracing the new understanding

of religion that had emerged in early Ming, which will be further explained below (Kojima 1996; Hamajima 2001; Wu 2016, pp. 103–5; Kong 2018, pp. 13–22).

The petition first called for the abolition of *wŏn'guje* (圓丘祭, lit. rite of round hill), a sacrificial ritual to the Lord of Heaven. Since the Chosŏn king was a foreign vassal from the Chinese viewpoint (see Han 2020, pp. 182–89), a perspective also adopted by Chosŏn intellectuals as their own, the petition demanded the removal of the improper practice of performing *wŏn'guje*, which only the Chinese emperor was eligible to perform. Cho Pak and his colleague officials also petitioned for the removal of Buddhist rites, Daoist rites, and others that had been designated state rites due to royal wishes during Koryŏ from the *sajŏn*. These rites were believed to grant the king's wishes through their spiritual efficacy. The petition further suggested that rites for various registered gods and local guardian deities should be performed in accordance with the Ming rite system, and proposed conducting state rites for Tan'gun (檀君 legendary figure in Korean history and mythology, regarded as the founder of the first Korean kingdom) and some Koryŏ kings who had been beneficial to the people.[12]

The petition for a comprehensive reform of the *sajŏn* was grounded in a changed understanding of religion.[13] During Koryŏ, a rite to a god typically meant an act of fulfilling people's wishes by relying on the god's spiritually potent power. In the new understanding, however, especially for state rites, a rite became an ethical deed. It should be carried out as an expression of rewards or thanks to the being (god) that granted benefits with its dutiful functions that it had carried on. Therefore, under this new perspective, the *sajŏn* register should be filled with the gods that had been beneficial to the state and its people, rather than the gods displaying remarkable spiritual efficacy.

In light of this new understanding of religion, state rites were no longer seen as acts to fulfill desires through gods' miraculous powers, as was common in the Koryŏ period. Instead, these rites became ethical acts of gratitude and repayment to gods who had conferred benefits upon the state and its people. For Cho Pak and his colleague officials, the *sajŏn*, which, in Koryŏ, often included gods known for their prominent miraculous powers, was now intended to list only those gods who had provided meritorious service. For proponents of this view, 'erroneous rites' (*ŭmsa* 淫祀), targeted for abolition, were considered rituals serving unlisted gods. Only gods with proven merits were to be registered in the *sajŏn*. In stark contrast to Koryŏ practices, rites performed for spiritually potent gods, particularly with the expectation of miraculous displays, were now viewed as erroneous.[14] For these officials, conducting a rite beyond one's status, or even for a meritorious god by an ineligible sponsor, was also deemed erroneous.[15]

Cho Pak and his colleague officials, guided by this new understanding, endeavored to reorganize the *sajŏn*. Their approach involved removing gods associated with spiritual efficacy, registering those known for their merits, and eliminating rites that exceeded one's status. The officials demanded the removal of all rites, including Buddhist and Daoist ones, that were registered in the *sajŏn* for their supposed miraculous powers to expel misfortune and bring fortune, aiming to eliminate the gods associated with spiritual efficacy from this state rite registry. The call to abolish the *wŏn'guje* was intended to prevent the performance of erroneous rites, as they violated the principles of Chosŏn, being a feudal state under Ming, even though they were not based on spiritual efficacy. Conversely, the proposal to conduct rites for Tan'gun, Kija 箕子, and some Koryŏ kings, who had achieved benefits for their people, aimed to add a proper rite (*chŏngsa* 正祀) to the *sajŏn*. If the petition of Cho Pak and his colleague officials had been accepted, the gods of spiritual efficacy would have been excluded from *sajŏn*, while the gods of merits would have been added. The rites beyond the status of a feudal state would also have been abolished.

Unlike officials in the Koryŏ period, these officials in Chosŏn did not view gods as anthropomorphic beings with spiritually potent powers. Instead, they saw gods as entities that had provided a meritorious service to the state and its people. They believed state rites should be carried out ethically and piously, as acts of gratitude and reward to these meritorious beings (gods). Consequently, they deviated from previous conventions[16] concerning

state rites for various gods and local guardian deities. Their efforts included changing the appellations of gods, such as stripping them of state-granted official ranks[17] and replacing statues with tablets. They also proposed that rites should be performed in every spring and fall, led by local magistrates, in accordance with the Ming ritual manual (*Hongwu lizhi* 洪武禮制).

Cho Pak and his colleague officials' suggestions were rooted in their distinguished understanding of religion. Their reform proposals were more a reflection of their acceptance of the new religious perspective of early Ming, rather than their own original ideas.[18] The petition by Cho Pak and his fellow officials was not unexpected at the time, given that the officials who led the foundation of Chosŏn aspired to emulate an ideal Chinese civilization in their own country (Park and Park 2023, pp. 179–85). They likely viewed the reform of the state rites as an integral part of their tasks for the newly established dynasty.

Early Ming policies, grounded in a new understanding of religion, essentially sought to reinforce traditional Chinese practices by abolishing what were considered barbarian customs. Although theoretically and ideologically sound, these policies often lacked practical applicability in local societies. At the individual level, people continued to adhere to their customary practices, such as performing rites to mountains, rivers, and heaven.[19] The Ming court endeavored to implement these reforms through laws and state rite manuals (Hamajima 1988). However, the failure of these policies to adequately reflect local and individual realities led to their gradual decline in effectiveness over time (Zhao and Yu 2003, pp. 129–31).

Given the context in Ming China, it is not surprising that the petition by Cho and others, which aligned with the ideological understanding of religion in early Ming, was largely overlooked. The petition presented an overly innovative approach that was out of step with not only the Chinese situation but also the realities in Korea. As a result, most of its demands, being too idealistic and in conflict with traditional religious understanding and practices, were either unenforceable or simply ignored.

For example, rites registered in the *sajŏn* for their spiritual efficacy continued to exist. In the case of state Buddhist rituals, only a few were abolished, with the political purpose of removing those associated with the authority and legitimacy of the previous dynasty (Kang 2017, pp. 491–92). The demands for changing the appellations of gods, such as stripping them of their official ranks and replacing statues with tablets, were not implemented, despite the royal decree.[20]

The fact that the petition of Cho Pak and his colleague officials was hardly implemented reflects the milieu of the period of King T'aejo and King Chŏngjong. During this period, any proposals that conflicted with conventional religious practices, especially those aimed at reforming the *sajŏn*, were generally not accepted, as the traditions of the Koryŏ era persisted. In fact, it is difficult to assert that there were numerous demands for changes to the *sajŏn*.

Meanwhile, after the foundation of Chosŏn, there was a move to limit ritual practices, allowing only ancestral rites. In October 1392, during the first year of King T'aejo's reign, Inspector-General Nam Chae 南在 and other officials proposed that rites should be acts of merit and virtue, and thus, should align with one's social status, prohibiting any erroneous rites that exceeded this limit.[21] This directive meant that officials and the public could not perform traditional rites for mountains and rivers, as these were beyond their eligibility; they were restricted to conducting only ancestral rites. Consequently, a royal decree was issued to forbid any rites that exceeded one's status and to encourage the establishment of *kamyo* (家廟 family shrines).[22]

However, this order was not properly observed by people. Reports indicated that people continued to conduct rites beyond their status, and although a series of similar orders were subsequently issued, they did not produce a clear outcome.[23] The persistent traditional understanding of religion, where people performed rites to various gods for fortune and disaster aversion, made it nearly impossible to adhere to the government's directive to perform rites solely for ancestors. As a result, during the reign of King T'aejo,

this order did not result in any significant change. Furthermore, under King Chǒngjong, the government took no further action to enforce these directives.

## 4. The Reign Periods of King T'aejong and King Sejong: Achievement of Government-Led Reform of the State Rites and Removal of Spiritual Efficacy

As King T'aejong ascended the throne, a significant shift occurred in the operation of the spiritual efficacy-based *sajǒn*, driven by his commitment to implementing an innovative understanding of religion. He actively led the removal of Buddhist and other erroneous rites from the state rite system, going so far as to address this issue on the day of his enthronement.[24] Such decisive actions by T'aejong were unprecedented. This approach was likely influenced by a more earnest pursuit of the ideal Chinese civilization during his reign, with T'aejong being a leading figure in this effort at the royal court.

As King T'aejong strove to eliminate erroneous rites from the state rite system, an initiative to abolish Buddhist rites was undertaken less than two months after his ascension to the throne.[25] This action specifically targeted state Buddhist rites.[26] During this period, the majority, if not all, of these rites were abolished. By the end of King T'aejong's reign, the scale of abolition had increased to the extent that only a few state Buddhist rites, such as the state *suryuk* (水陸 water–land) ritual, remained ([Kang 2017](#), pp. 498–509). Given that the abolished state Buddhist rituals were primarily conducted for kings to fulfill their wishes through spiritual efficacy, this abolition likely represented a part of T'aejong's project to eliminate the element of spiritual efficacy from the state rite system.

Although the reform of the *sajǒn*, based on the new understanding of religion, was not implemented concurrently with the abolition of the state Buddhist rites in the year of King T'aejong's enthronement, it steadily progressed with the subsequent removal of erroneous rites.[27] During King T'aejong's reign, the *sajǒn* reform generally aimed to enforce most of the previous policies that had failed under King T'aejo and Chǒngjong, largely due to the conventional spiritual efficacy-oriented understanding of religion and religious practice. Additionally, new policies were also introduced for this purpose. King T'aejong's reign marked a pivotal period in establishing this innovative approach to religion. However, as the *sajǒn* reform confronted and sought to supplant traditional religious understanding and practice, its implementation was not smooth, with many actions not being firmly enforced ([Ch'oe 2020](#), pp. 28–37).

The reform of the *sajǒn* during King T'aejong's reign, aimed at moving beyond the spiritual efficacy-based state rite system, involved various actions, which can be categorized into five groups. Firstly, efforts were made to align the state rites with the innovative ritual system of early Ming. Although the wholesale reform of the state rite system, in accordance with the *Hongwu lizhi*, was initiated in King T'aejo's reign, it was not actively implemented until King T'aejong's time.[28] This delay stemmed from the challenges in accepting and enforcing a ritual system vastly different from the previous one. With King T'aejong and most of his court officials firmly committed to adopting the innovative ritual system of early Ming, the operation of the state rite system following the *Hongwu lizhi* became achievable. This included changing the appellation of gods, such as abolishing their official titles and replacing statues with tablets. However, as evidenced by the project's continuation beyond T'aejong's reign, it required time and persistent effort to reach completion ([Yi 2009](#), pp. 101–2).

Secondly, some actions were aimed at removing various shrines dedicated to city gods (*sǒnghwang* 城隍) or mountain–river gods, which had been registered as places of spiritual efficacy in the *sajǒn*. These measures most dramatically illustrated the efforts to eliminate the element of spiritual efficacy from the state rite system. Notably, the removal decision made in the ninth year of King T'aejong's reign (1409)[29] marked the initial and a substantial step towards removing sites associated with spiritual efficacy from the *sajǒn*. This move symbolized the start of a project to refill the *sajǒn* with deities recognized for their merits to the state and its populace, signifying a shift in religious concept and a departure from the practices based on spiritual efficacy that were pervasive in the previous dynasty of

Koryŏ period. Throughout King T'aejong's tenure, albeit on a smaller scale, further efforts were made to expunge shrines reliant on spiritual efficacy from the *sajŏn*. Although the specific names of the removed shrines are not recorded, the general trend during T'aejong's era suggests that most shrines reliant on spiritual potency were eliminated from the *sajŏn*, implying that those recorded during the Koryŏ period for their spiritual efficacy were likely removed from the *sajŏn* at this time. Meanwhile, the tasks unfinished during T'aejong's reign were continued and concluded in the era of King Sejong, marking the culmination of this transformative initiative.

Thirdly, the actions taken during King T'aejong's reign included the classification of mountains and rivers within the *sajŏn*.[30] Previously, the rites for mountains and rivers registered in the *sajŏn* were not categorized, showcasing a different approach compared to China. This absence of classification can be attributed to the selection and registration in the *sajŏn* based on spiritual efficacy. At that time, the act of registering these spiritually potent mountains and rivers in the *sajŏn* was regarded as significant. Moreover, performing state rites there and praying for fortune, alongside seeking protection from misfortune through the spiritual efficacy of these mountains and rivers, was considered important. For these reasons, contrary to the practices in China, the government had little interest in classifying them based on their merits. However, after the removal of many places of spiritual efficacy from the *sajŏn* during King T'aejong's reign, the government began efforts to categorize the remaining mountains and rivers in the *sajŏn* based on their merits ([Yi 2009](#), p. 95).

Fourthly, during King T'aejong's reign, actions were taken to abolish *wŏn'guje*, which was regarded as an erroneous rite beyond the Chosŏn king's prerogative as a vassal. Initially, the *wŏn'guje* was temporarily abolished during the reign of King T'aejo following a petition by Cho Pak and his fellow officials. However, it was reinstated about two years later when the king accepted a petition from the ministry of rites, which argued that what had been practiced for a long time could not be lightly abolished.[31] In essence, its reinstatement signifies the difficulty of abandoning conventional practices. Yet, in 1412, during the twelfth year of King T'aejong's reign, *wŏn'guje* was once again abolished. King T'aejong firmly adhered to the ritual principle that the heavenly son should perform rites to heaven and earth, while a vassal should conduct rites to the mountains and rivers within his jurisdiction.[32] While *wŏn'guje* was sporadically conducted even after 1412, primarily for praying for rain, it differed from its previous regular practice. Before 1412, it was regularly held in January and April, and occasionally during severe droughts. After 1412, however, the regular, full-scale *wŏn'guje* was not resumed, and only the irregular form was occasionally performed. The government did not view the irregular *wŏn'guje* ceremony as a violation of the principle, understanding that even a feudal state could temporarily perform the rite in times of severe drought.[33]

Finally, during King T'aejong's reign, actions were taken to adopt and implement various rites from the *Hongwu lizhi*. This included the rites for wind, clouds, thunder, rain, mountains, rivers, and guardian deities, the *yŏje* 厲祭 rite, and the rite for the local gods of soil and five grains (*sajikche* 社稷祭).[34] Originally established during early Ming period, these rites emerged from an innovative understanding of religion. Therefore, their acceptance in King T'aejong's reign signified more than just incorporating certain rituals from the *Hongwu lizhi*; it represented the actualization of a new religious vision. These actions were indicative of the intellectual and ideological readiness of the time to embrace such ritual changes.

The efforts to reform the state rite system during King T'aejong's reign, inspired by a new understanding of religion and as a part of the movement to embody an ideal Chinese civilization in Korea, not only continued but also expanded during King Sejong's reign. This period witnessed a more comprehensive execution of some actions that had previously not been well enforced, further advancing the reforms in Chosŏn. During the reign of King Sejong, the notion that the irregular *wŏn'guje* could be performed during times of drought was rejected. The occasional practice of the irregular *wŏn'guje*, following the abolition of its regular form, likely persisted due to a social milieu where conventional practices

and religious understanding were still influential among government officials. However, this practice was definitively banned during King Sejong's rule. In 1442 (the twenty-fifth year of King Sejong's reign), arguments supporting the emergency performance of *wŏn'guje* were refuted based on the principles of propriety for a feudal lord, and the king concurred that even such irregular instances of *wŏn'guje* should not occur.[35] Subsequently, King Sejong declined officials' requests to perform *wŏn'guje* solely as a prayer for rain. After 1449 (the thirty-first year of King Sejong's reign), requests for this rite ceased entirely.[36]

After the regular *wŏn'guje* was abolished in 1412, during the twelfth year of King T'aejong's reign, the irregular form of *wŏn'guje* was subsequently abolished under King Sejong. Even the instances of *wŏn'guje* held in emergency situations were not exempt from criticism, as they, too, violated the ritual principles and were considered erroneous rites beyond the status of Chosŏn. This context makes it clear that the actions taken during King Sejong's reign aimed to align state rites more firmly and strictly with the proprieties befitting a feudal state ([Han 2002]).

As previously mentioned, the substantial efforts undertaken during King T'aejong's reign to align the operation of state rites with the innovative ritual system of early Ming required more time and effort to produce more evident outcomes, which were achieved during the reign of King Sejong. In the second month of the sixth year of his reign, King Sejong addressed concerns regarding local guardian deities and mountain gods being designated titles like '*taewang*' 太王 or '*taehu*' 太后. He highlighted issues such as improper rites being conducted at mountain shrines, contrary to the old regulation that altars should be set at the mountains' feet, and rites to mountains and rivers (guardian deities) being improperly performed by the general populace, exceeding their status. Consequently, King Sejong instructed Yi Chik 李稷 and three other officials to examine the old regulations related to the granting of titles and the establishment of shrines. He expressed his point of view that altars should be established at the base of mountains, tablets for the gods should bear the inscription of 'god of a certain mountain', and only state rites should be conducted to avoid the erroneous performance of rites.[37]

The practices that King Sejong identified as problematic were entrenched conventions from the Koryŏ period. These likely originated from the tendency to view gods as anthropomorphic beings with spiritually potent powers, with rites to these gods seen as means to fulfill desires by relying on that power. King Sejong's proposed solutions can be viewed as an extension of King T'aejong's reforms, with the addition of establishing altars at the base of mountains and accompanying them with tablets. Despite King T'aejong's efforts, it appears that these older practices continued. While King Sejong referenced the old regulations, his approach was grounded in the new understanding of religion that had emerged in early Ming.

Contrary to King Sejong's aspirations, four officials, including Yi Chik, expressed their opinion that the practices identified by the king, which had long been conducted in both China and Korea, could be continued without issue.[38] Had this viewpoint been accepted, it would have nullified all the extensive efforts put into the wholesale reform of the *sajŏn* and its alignment with the Ming ritual system, effectively undoing the progress achieved during King Sejong's reign.

However, the opinion of the four officials could not reverse the trend established during King T'aejong's reign, as the world had already undergone significant changes. Rather, afterwards, the situation transpired in a way that would fit King Sejong's vision. Such changes persisted even though "establishing an altar at the foot of the mountains" was partially enforced. In the eighth month of the twelfth year of King Sejong's reign, he accepted the ministry of rite's proposal to reform the shrines of mountains and rivers, which were still adhering to old practices. This proposal was based on a patrolling magistrate's report on the altars and shrines of mountains and rivers in the *sajŏn*. The magistrate's report[39] aimed at implementing reforms, based on King T'aejong's directive to the Minister of Rites, issued in the sixth month of the thirteenth year of his reign. The directive instructed that statues of mountain–river gods and guardian deities should be replaced

with tablets mounted on wooden plaques, thereby rectifying the *sajŏn* and discarding old conventions from the Koryŏ period.[40] The initiative for this reform began during King T'aejong's reign[41] and was enforced again under King Sejong, as many altars and shrines in the *sajŏn* had yet to fully abandon the old conventions. Therefore, it represented a continuation of the actions initiated in King T'aejong's time. In the third month of the nineteenth year of King Sejong's reign, the government approved most of the plans proposed by the Ministry of Rites. This proposal was based on reports from patrolling magistrates. It detailed the system of the altars, tablets, and shrines for the mountains, streams, and rivers.[42] The Ministry's plan could be seen as the endpoint of the *sajŏn* reform process that had been underway up to that point.

After the actions taken in the third month of the nineteenth year of King Sejong's reign, there were no further actions required to reinforce or supplement the previous measures, such as changing appellations, establishing tablets, and removing statues, which were all aimed at reforming the old conventional practices. These specific issues no longer posed problems for the reform of the *sajŏn*. Similarly, no additional actions were necessary for the exclusion of places of spiritual efficacy from the *sajŏn* or the classification of mountains and rivers within it. This lack of further actions indicates the successful completion of this reform project (Ch'oe 2009b, pp. 239–42).

The reform of the *sajŏn*, which embraced a new understanding of religion and moved away from the spiritual efficacy-oriented approach, was actually completed during King Sejong's reign.[43] Its outcome was reflected in the five rites in the *Sejong sillok* 世宗實錄.[44] The state rites in the five rites continued with little alteration throughout the Chosŏn dynasty. This continuity demonstrates that the efforts to reform state rites and eliminate the element of spiritual efficacy were successfully concluded during the reigns of Kings T'aejong and Sejong. Furthermore, the results of these reforms laid the foundation for the practice of state rites throughout Chosŏn period.

## 5. The Modes of Use of Spiritual Efficacy in State Rites and Its Features after the Reign of King T'aejong

With the reign of King T'aejong marking a significant turning point, the state Buddhist rites, along with other spiritual efficacy-based state rites, were removed from the *sajŏn*. Only deities acknowledged for their merits to the state and its people were registered, reflecting the new understanding of religion. State rites were thus re-envisioned as ethical acts, performed as expressions of gratitude and recognition to the gods who had bestowed benefits upon the state and its people.

However, this does not imply that Buddhist rituals and other rites outside the state rite system were completely abandoned by the government after the reign of King T'aejong. In fact, it was not uncommon for erroneous rites, including Buddhist rites, to be carried out.[45] These rites were often conducted as a last resort in emergency situations deemed beyond human control, such as natural disasters, epidemics, or deaths in the royal family, despite being considered improper (Ch'oe 2016, pp. 400–3).

It is noteworthy that even after King T'aejong's reign, while erroneous rites continued to be performed at the state level, the understanding behind them had significantly changed. Even though the king (state) regarded these rites as improper or undesirable, they were acknowledged as necessary in certain dire situations where no alternative was available. For instance, in cases of uncontrollable disease or death, King T'aejong, although considering it improper, resorted to performing a Buddhist healing rite, justified by the 'mind which cannot bear to see the suffering of others' 不忍之心 as a Confucian rationale.[46] In essence, conducting a Buddhist rite was seen as an expedient means, albeit not appropriate.

Particularly noteworthy is that rites considered erroneous were still relatively frequently conducted to dispel misfortunes. For instance, during droughts, monks were summoned to conduct rain prayers, or irregular rain rites were also performed in places of spiritual efficacy that were not registered in *sajŏn*.[47] While on the surface, these rites seemed to address disasters through spiritual efficacy, similar to practices in the Koryŏ period, they

were actually carried out against a backdrop of a different ideological standpoint distinct from that of Koryŏ.

For example, King T'aejong once posed the following query to his officials: "From old days, droughts and floods are attributed to the king's lack of morality. While monks and shamans are mobilized to pray for rain, I am uncomfortable with this. Even if it rains, it isn't likely due to their power … I think it is right to cease this prayer and rite and to govern properly … What are your thoughts on seeking heaven's compassion through such erroneous means?" In response, Kim Yŏji 金汝知 said "Though conducting such a rite to gods is not the way of the ancient sage-kings, for trying to solve problems praying to any deity in situations of disaster where people are suffering is an old practice. … It would be better to carry out these prayers quietly." The king agreed with this suggestion.[48] At that time, even though rain rites were conducted with monks and shamans, as in Koryŏ, there was a shared belief among the king and his officials that disasters resulted from the king's immorality and could be resolved through the king's self-reflection, moral cultivation with fear of heaven, and correction of wrong affairs. It was also assumed that rain during such rites was not a result of the ritual itself. The continuation of rain rites with monks and shamans, regarded as erroneous, stemmed from the belief that a rite should 'be performed for any deity to solve problems in situations of disaster where people are suffering' 靡神不舉 (Ch'oe 2016, pp. 400–3). Such cases were not uncommon even after King T'aejong's reign.[49]

## 6. The Attempt to Reform Personal Rites and Its Failure during the Reign Periods from King T'aejong to King Sŏngjong

While the spiritual efficacy-based *sajŏn* system was reformed during the reign of King T'aejong, there were not many changes in rites on an individual level. Not long after ascending the throne, King T'aejong raised a question to his officials during the royal lecture (*kyŏngyŏn* 經筵) about abolishing an erroneous rite at an individual level. While he firmly believed in the necessity of the ban, even initiating the discussion himself, and was aware that many of his officials agreed with him, the king still hesitated to enforce the ban, concerned about potential resistance from the general populace.[50] This apprehension led to the stagnation of the reform proposal stating that people should only perform rites for their ancestors (Ch'oe 2020, pp. 37–38).

Immediately following the king's leadership in abolishing *wŏn'guje*, in line with the principle of status-appropriate rites,[51] calls emerged once again to ban conducting rites like mountain–river rites on an individual level, with the exception of ancestral rites. In October 1412, the twelfth year of King T'aejong's reign, the *Saganwŏn* (司諫院 Office of the Censor General) submitted a petition to strictly prohibit the performance of mountain–river rites and other erroneous rites, citing their improper conduct.[52] The petition essentially advocated for a regulation stating that only ancestral rites could be conducted. About a month later, in the eleventh month of the same year, King T'aejong protested that people were conducting mountain–river rites beyond their status, as well as Buddhist death rites for the deceased, and ordered the *ŭijŏngbu* (議政府 State Council) to discuss the *Saganwŏn*'s petition.[53] During the deliberations, while many officials in the *ŭijŏngbu* voiced support for abolition, some opposed it, arguing that Buddhist and other long-established rites should not be abruptly abolished. Due to this opposition, the discussion on abolition did not advance further.[54]

In the end, the attempt to correct the conventional practice of people performing mountain–river rites beyond their status failed. A month after this, in the twelfth month of the twelfth year of King T'aejong's reign, the king ordered the *sahŏnbu* (司憲府 office of the inspector general) to ban people from praying for their fortune in the shrine for the guardian deity of Mt. *Songak* 松嶽.[55] It was an action to prevent too many people from gathering and performing a rite there, not to ban the performance itself of the mountain–river rite by people. However, the ban did not even last long. In the first month of the eighteenth year of King T'aejong's reign, the king ordered that the rite for Mt. *Songak* and

Mt. *Kamak* 紺岳 be allowed.[56] Even the ban against some of the mountain–river rites that became problematic was not kept. This indicates that it was a very difficult or even near impossible task to restrict a mountain–river rite that people had long performed, designating it improper with the new vision of religion.

In this context, the *isa* 里社, a village rite dedicated to the god of the soil and the god of the five grains, was instituted in the fourteenth year of King T'aejong's reign (1414). Recorded in the *Hongwu lizhi*, the *isa* was intended as a long-term strategy to phase out erroneous rites, such as the people's mountain–river rite, by introducing a local communal rite that adhered to ritual propriety, aside from the ancestral rites. However, *isa*, being part of early Ming's broader program to impose social order in local communities through Confucian laws and regulations ([Bol 2010](), pp. 256–60), likely faced challenges in gaining widespread acceptance. Despite its enforcement and actual practice in some areas, the *isa* struggled to resonate with ordinary people, primarily because it did not align with their longstanding religious beliefs and practices.

Reforming individual-level rites, in contrast to the state rites system (*sajŏn*), was likely to be successful only if the new religious understanding spread more widely and deeply among the people. Consequently, this process was undertaken more gradually, resulting in fewer noticeable outcomes during King T'aejong's reign. Even after King T'aejong's reign, the situation with religious practices remained largely unchanged. Despite establishing a legal foundation for a comprehensive ban against erroneous rites during the later period of King Sejong's reign,[57] this state ban was eventually withdrawn in King Sejo 世祖's reign period,[58] rendering previous reform efforts largely ineffective. The ban was reintroduced under King Yejong 睿宗[59] and further strengthened during King Sŏngjong 成宗's reign. However, its effectiveness was significantly reduced compared to the period of King Sejong.[60] The scope of the ban was limited, applying primarily within the capital city (Seoul) and targeting only the women of the Confucian literati class. Subsequently, no additional legal or institutional measures were taken to reinforce the ban. With the continued practice of the spiritual efficacy-based mountain–river rite among the populace, the enforcement of the ban on erroneous rites was implemented in a restricted manner, and even the existing prohibitions were not fully executed ([Ch'oe 2009a](), pp. 207–11).

While the reform of the *sajŏn* could be enforced in a top-down manner with the government's firm resolve, reforming individual-level rites faced challenges due to the lack of widespread acceptance of the new religious understandings among the people. Enforcing a ban that allowed only ancestral rites would likely have provoked significant resistance and negative social consequences. In this context, the government refrained from strictly prohibiting the traditional practice of performing mountain–river rites beyond one's status, not expecting immediate compliance. Instead, the intention was probably to facilitate a gradual change in people's consciousness and practice over the long term.[61] This strategic approach likely contributed to the continued prevalence of traditional practices without significant impediments.

## 7. Afterword: Change in the Realm of Personal Rituals in the Late Chosŏn Period

Influenced by newly emerged religious concepts in the early Ming dynasty, the government in the early Chosŏn dynasty sought to reform both state and personal rituals. This reform effort was aimed at eliminating the elements of spiritual efficacy and ensuring these rites were in harmony with the newly introduced religious paradigms of the period. Consequently, the government managed to successfully reform state rites by eliminating the spiritually potent components, thereby realigning them with the Neo-Confucian religious vision. In contrast, despite the aim of promoting the exclusive practice of conducting ancestral rites in a Confucian manner, attempts to modify the personal religious practices of officials and the broader populace, which involved reliance on deities believed to possess supernatural powers, failed to meet the intended objectives.

The general populace continued to pursue supernatural interventions through erroneous rites (*ŭmsa*), thereby necessitating limited enforcement against such practices. While

the government could decisively implement ritual changes in a top-down manner for state ceremonies, altering personal-level rituals encountered significant resistance due to a general lack of acceptance of the new religious concepts among the people. Enforcing a policy that solely permitted ancestral rites was not only impractical, but also likely to elicit substantial resistance and negative outcomes. Given this context, the government might have aimed for a gradual shift in cultural norms and attitudes rather than the direct prohibition of established practices.

By the late Chosŏn period, there were indications of innovation in personal religious practices. The mid-17th century represents a crucial juncture where the prevalence of erroneous rites (*ŭmsa*) began to decline. This change was not simply a result of stronger legal and institutional governance or more assertive indoctrination by the elite, but was primarily due to the populace's own adoption and internalization of Neo-Confucian moral standards. As new religious ideas spread within communities and traditional religious fervor and practices diminished, the local populace's enthusiasm for engaging in forbidden rituals lessened, and opposition to efforts aimed at banning such practices decreased. This development in the mid-17th century suggests that changes in personal sacrificial practices became firmly established in the late Chosŏn period.

**Funding:** This research received no external funding.

**Conflicts of Interest:** The author declares no conflict of interest.

## Notes

1. In this paper, the meanings of 'state rites' and 'personal rites' are as follows: 'state rites' refer to public rites conducted by the state (king), whereas 'personal rites' refer to private rites carried out by individuals.

2. Within this study, 'spiritual efficacy' is defined as the ability of religious rituals, objects, or places to produce desired spiritual effects or outcomes, as employed by Valerie Hansen in her research on religious practices and beliefs in Chinese history. See Hansen (1990).

3. The merits attributed to the mountain and river gods are understood to be the benefits they bestow through their natural functions, such as generating clouds and rains or aiding the growth of crops. These functions are what the mountain and river gods performed, not acts they could willfully execute in response to human rituals or the sincerity of those conducting them. Instead, these actions were regarded as inherent duties of these deities, which contributed to the wellbeing of the state and its people. See Chŏng Tojŏn 鄭道傳 (1342~1398). *Chosŏn kyŏngguk chŏn* 朝鮮經國典 sang 上, Yejon 禮典 Chesin sajŏn 諸神祀典.

4. Neo-Confucianism is a revitalization of Confucian thought that emerged in Song Dynasty China, emphasizing moral virtues, self-cultivation, and a philosophical understanding of the universe through the principles of '*Li*' (principle) and '*Qi*' (vital energy).

5. Ahn (2018) can be referenced on this matter.

6. Silla 新羅, adopting the Tang system, also categorized its state rites into great, medium, and small rites. However, only some of its state rites were classified under these categories. Interestingly, all rites within these categories were mountain–river rites (*Samguk sagi* 三國史記 32, chapchi 雜志 1, chesa silla 祭祀 新羅). Additionally, several state rites that did not fit into these categories, including some large-scale rites like the chongmyo rite, were also practiced. For more details, see Chae (2008).

7. The Confucian officials of Chosŏn, responsible for compiling the *Koryŏsa* 高麗史, deemed it appropriate to organize state rites into the categories of great, medium, and small rites. They classified those Koryŏ period rites that did not align with these categories as *chapsa* (miscellaneous rites).

8. For more on Koryŏ *chapsa* 雜祀, see Kim (2007).

9. *Koryŏsa* 高麗史 99, Yŏlchŏn 列傳 12, Ham Yuil 咸有一.

10. Only a small number of local guardian deities, like those of *Tŭngju*, were actually registered in the *sajŏn*, likely because these deities were considered to possess significant spiritually potent powers. For a detailed analysis of the biography of Ham Yuil in the *Koryŏsa*, see Ch'oe (2017, pp. 317–22).

11. *T'aejo sillok* 太祖實錄 1, kyŏngsin of the eighth month of the first year of King T'aejo.

12. Chŏng Tojŏn, aligning with a similar ideological stance, advocated for the inclusion of beings with merits in the *sajŏn*. He asserted that those listed in the *sajŏn* were deities who had demonstrated their merits to humanity (凡載祀典者 皆有功德於民), as described in the *Chosŏn kyŏngguk chŏn*. These included not only gods of nature like mountains and rivers, but also historical figures whose heroic deeds benefitted the people, as detailed in the *Chosŏn kyŏngguk chŏn* sang, Yejŏn Chesin sajŏn. For instance, figures such as *Tan'gun* 檀君, the first monarch to receive the heavenly principle in Korean history, and *Kija* 箕子, known for enlightening the Korean people, were deemed worthy of inclusion in the *sajŏn*.

13    This part is based on Ch'oe (2020).

14    *Chŏngsa* 正祀 (proper rites), as opposed to the erroneous rites, referred to rites to the gods with merits in the *sajŏn*.

15    During the Koryŏ period, rites considered erroneous were often associated with an excessive belief in shamanism, especially in a context where shamanistic practices had become widely popular. It was not the shaman rites themselves that were deemed incorrect, but the excessive influence and prevalence of shamanistic rituals. Consequently, the strategy to eliminate these erroneous rites involved expelling shamans as a means to diminish the pervasive impact of shamanistic practices (Ch'oe 2009a).

16    For a deeper understanding of the previous practices, see Yi (1998, pp. 126–33).

17    This reform regarding the appellation of deities was based on a newfound understanding that the Tang dynasty's practice of granting official titles to mountain, river, and guardian deities was in conflict with ancient ritual norms and Confucian principles (*Koryŏsa* 42, seventh month of the nineteenth year of King Kongmin).

18    The religious understanding that emerged in the early Ming period, adopted by Cho Pak and his colleague officials, can be found in the following materials: *Ming Taizu shilu* 明太祖實錄 35, bingzi in the tenth month of the year of enthronement of Hongwu 洪武; *Ming Taizu shilu* 53, guihai in the sixth month of the third year of Hongwu; and *Ming Taizu shilu* 53, jiazi in the sixth month of the third year of Hongwu. For insights into the erroneous rites in early Ming, see Zhao and Yu (2003, pp. 127–29); Luo (2006, pp. 131–34).

19    *Ming Taizu shilu* 53, jiazi in the sixth month of the third year of Hongwu.

20    *T'aejong sillok* 太宗實錄 25, ŭlmyo of the sixth month of the thirteenth year of T'aejong.

21    *T'aejo sillok* 2, kihae of the ninth month of the first year of T'aejo.

22    *T'aejo sillok* 2, imin of the ninth month of the first year of T'aejo.

23    *T'aejo sillok* 8, kabin of the twelfth month of the fourth year of T'aejo; *T'aejo sillok* 8, muo of the twelfth month of the fourth year of T'aejo; *T'aejo sillok* 11, chŏngmi of the fourth month of the sixth year of T'aejo.

24    *Chŏngjong sillok* 定宗實錄 6, kyeyu of the eleventh month of the second year of Chŏngjong.

25    *Chŏngjong sillok* 6, imja of the twelfth month of the second year of Chŏngjong.

26    *T'aejong sillok* 1, chŏngch'uk of the first month of the first year of T'aejong.

27    *T'aejong sillok* 1, sinmi of the fourth month of the first year of T'aejong.

28    *T'aejong sillok* 25, ŭlmy of the sixth month of the thirteenth year of T'aejong.

29    *Sejong sillok* 世宗實錄 29, the seventh month of the seventh year of Sejong; *Sejong sillok* 46, kyech'uk of the eleventh month of the eleventh year of Sejong.

30    *T'aejong sillok* 25, ŭlmyo of the sixth month of the thirteenth year of T'aejong; *T'aejong sillok* 28, sinyu of the eighth month of the fourteenth year of T'aejong.

31    *T'aejo sillok* 6, muja of the eighth month of the third year of T'aejo.

32    *T'aejong sillok* 24, chŏngch'uk of the eighth month of the twelfth year of T'aejong; *T'aejong sillok* 24, kyŏngjin of the eighth month of the twelfth year of T'aejong.

33    For more details on the abolition of *wŏn'guje* during the reign of T'aejong and its temporary reinstatement, see Ch'oe (2013, pp. 62–71).

34    *T'aejong sillok* 7, muin of the sixth month of the fourth year of T'aejong; *T'aejong sillok* 11, kyehae of the sixth month of the sixth year of T'aejong; *T'aejong sillok* 21, mujin of the fifth month of the eleventh year of T'aejong.

35    *Sejong sillok* 101, kyehae of the seventh month of the twenty-fifth year of Sejong.

36    *Sejong sillok* 01, ŭlch'uk of the seventh month of the twenty-fifth year of Sejong; *Sejong sillok* 105, chŏngmyo of the seventh month of the twenty-sixth year of Sejong; *Sejong sillok* 125, imo of the seventh month of the thirty-first year of Sejong.

37    *Sejong sillok* 23, chŏngsa of the second month of the sixth year of Sejong.

38    *Sejong sillok* 23, chŏngsa of the second month of the sixth year of Sejong.

39    *Sejong sillok* 49, kapsul of the eighth month of the twelfth year of Sejong.

40    In the '*T'aejong Sillok*' (*The Annals of King T'aejong*), volume 25, under the date ŭlmyo of the sixth month of the thirteenth year of T'aejong's reign, there is a detailed account of the royal order issued to remove the official titles of deities and to change their appellations to a standardized format, such as 'the guardian deity of a certain county' or 'the god of a certain sea, mountain, or river'.

41    *T'aejong sillok* 25, ŭlmyo of the sixth month of the thirteenth year of T'aejong 13.

42    *Sejong sillok* 76, kyemyo of the third month of the nineteenth year of Sejong.

43    The policy of King Sejong with respect to state Buddhist rites followed that of King T'aejong, with efforts to remove these rites from the state rite system becoming more concrete and established, as detailed in Ch'oe (2019, pp. 26–29).

44    *Sejong sillok Oryeŭi killye ŭisik*.

45    They were no longer held as official state rites.

[46]　*T'aejong sillok* 15, chŏngch'uk of the first month of the eighth year of T'aejong.

[47]　*T'aejong sillok* 26, imo of the seventh month of the thirteenth year of T'aejong; *Sejong sillok* 124, sinhae of the sixth month of the thirty-first year of Sejong.

[48]　*T'aejong sillok* 26, imo of the seventh month of the thirteenth year of T'aejong.

[49]　After the reign of T'aejong, rites based on spiritual efficacy were conducted not out of a belief in their manifest spiritual effects, but rather, from a sense of obligation to do whatever possible for the people in situations where they were suffering. To give a few examples, *T'aejong sillok* 22, kyŏngo of the seventh month of the eleventh year of T'aejong; *T'aejong sillok* 26, imo of the seventh month of the thirteenth year of T'aejong; *Chungjong sillok* 95, kyech'uk of the fifth month of the thirty-sixth year of Chungjong.

[50]　*Chŏngjong sillok* 6, musin of the twelfth month of the second year of Chŏngjong.

[51]　*T'aejong sillok* 24, kyŏngjin of the eighth month of the twelfth year of T'aejong.

[52]　*T'aejong sillok* 24, kyŏngsin of the tenth month of the twelfth year of T'aejong.

[53]　The petition of saganwŏn also raised the issue of Buddhist rites.

[54]　*T'aejong sillok* 24, ŭlsa of the eleventh month of the twelfth year of T'aejong.

[55]　*T'aejong sillok* 24, sinmi of the twelfth month of the twelfth year of T'aejong.

[56]　*T'aejong sillok* 35, ŭlhae of the first month of the eighteenth year of T'aejong.

[57]　*Sejong sillok* 101, chŏngmi of the eighth month of the twenty-fifth year of Sejong; *Sejong sillok* 101, kyech'uk of the ninth month of the twenty-fifth year of Sejong.

[58]　*Sejo sillok* 世祖實錄 4, ŭlhae of the fifth year of the second year of Sejo.

[59]　*Yejong Sillok* 睿宗實錄 3, pyŏngsin of the second month of the first year of Yejong; *YejongSillok* 7, kapsin of the ninth month of the first year of Yejong.

[60]　*Sŏngjong Sillok* 成宗實錄 14, sinch'uk of the first month of the third year of Sŏngjong; *Kyŏngguk taejŏn* 經國大典 5, hyŏngjŏn kŭmje.

[61]　*T'aejong sillok* 35, ŭlhae of the first month of the eighteenth year of T'aejong.

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
