# Peer review of "The Efforts of Government-Driven Reform of Both State and Personal Rites in Early Chosŏn: A Historical Shift from Spiritual Efficacy to Meritorious Practice"

_religions, doi:10.3390/rel15040418_

Round 1

Reviewer 1 Report

Comments and Suggestions for Authors

Peer Review of "Efforts of Government-led Reform"

The paper presents a cogent and interesting overview of ritual reform in the early Chosŏn dynasty; it also introduces important issues in the history of religion to a broader audience. However, the presentation is too haphazard. The author assumes too much prior knowledge on the part of the reader, esp. regarding the 1) the basic contours of Korean history 2) ideas in East Asian religion 3) facets of Korean society, 4) rationale for specific translations of Korean language terminology and other technical terms (e.g. ŭmsa), among other things. The author also does not engage with relevant scholarship to explain why this argument is meaningful (which I believe it is) and does not engage in English language literature, either about religion in general or the history of Korean religion. I might suggest looking at recent work by Juhn Ahn or Gregory Evon, or Maya Stiller, or even older work by Martina Deuchler for a point of reference. It doesn't need to be an extensive engagement, but some linkage to English-language work would help the author situate the work for the English-language readership. A brief mention or citation would suffice. The author does engage with Korean language scholarship, but doesn't make explicit how this paper enters into the debate or the state of the field in Korean religious studies.

In terms of presentation, the author also relies heavily on passive voice, which obfuscates the argument, the narrative, and the actors in question. The English prose needs to be revised as well, mostly for clarity. The author also italicizes people's names, which is unconventional.

The author also needs to have a clearer introduction, as well as a conclusion to the article. The key term, "wonder" is also poorly defined and is not situated in its historical/literary/linguistic context. What does the author mean by wonder?

I believe if the author fixes these issues of presentation and convention and makes the intervention more explicit, it will stand to be a valuable contribution to the English language literature on Korean religion. Someone in the field who is familiar with the subject matter will be able to intuit the author's desired contribution, but as it stands it is just too difficult to follow—it is also difficult for the reviewer to evaluate the scholarly contribution on its own merits because of clarity-related issues. The author should work on improving the presentation so the work can reach a wider audience.

    •  

Comments on the Quality of English Language

I suspect the paper might have been translated directly from Korean, which may explain some of the usage/convention-related issues; the author might consider working with a colleague or editor to help fix the draft.

I have given some examples of areas where the author could add more information or revise to increase the clarity of the article:

  • pg. 1

    • "how the government"; government of what?

    • Need more background information about what is Chosŏn, what is Koryŏ (dates? time? place?); not clear to reader without prior knowlege of Korean history

    • Does it make sense to make such a stark difference between Koryŏ and Chosŏn? Can we be more specific?

    • "there was a demand"-> from whom?

    • "was emphasized" -> by whom?

    • "enforcing these reforms" -> by whom?

    • "new religious perception"-> which is what?

    • "on a different understanding of religion" -> what difference? This needs to be spelled out succinctly and directly

    • "This reform"-> which reform? What period? By whom?

    • "was conducted"

      • Generally speaking, the author needs to get rid of the passive voice and be more explicit about who is doing what.

  • pg 2

    • "It is also understood"

      • By whom?

    • "These perspectives"

      • Whose perspectives? No citation here.

    • "they hardly pay attention"

      • Who is they?

    • "the various issues that could have failed it"

      • Which issues?

    • So far the introduction is extremely vague. It needs to be rewritten to be clearer and more direct about the author's specific claims. There should also be a succinct 1-2 sentence statement of the author's thesis. The abstract contains some of this language and it can be brought back here.

  • pg. 2, part 2

    • "operation of the rites, based on wonder"

      • What the author means by wonder should be defined clearly and explained in the introduction

    • Background information, 3 paragraphs

      • This background is useful, but the author should explain why this background is being offered.

    • "did not display wonder"

      • What is the classical Chinese term used here? What does this term mean in the context of East Asian religion?

    • "wonder of the gods"

      • What does this mean?

      • Author should cite relevant literature in religious studies to explain what is wonder.

    • Ham

      • Why is his name italicized?

    • "It is presumed"

      • By whom?

  • pg 3/ part 3

    • regin -> reign

    • new dynasty- no context for reader

    • "technically a feudal lord from the Chinese viewpoint"

      • Need citation or explanation; cannot assume audience will be familiar with this concept

    • Tan'gun

      • No context

    • I stop here for detailed comments; there are similar issues throughout the paper.

  • Bibliography:

    • The McCune Reischauer romanization and citation style is generally applied correctly (some small issues that should be revised on resubmission); one issue is Korean and Chinese names should be given as: Han Ugŭn (no comma), as opposed to Han, Ugŭn.

    •  

Author Response

Dear a reviewer,

Thank you for the opportunity to revise my manuscript (Manuscript ID: religions-2851917) following your comments. I have thoroughly reviewed each comment and made careful revisions to my manuscript accordingly. Below, I provide a detailed response to each of your' suggestions and comments.

I appreciate the constructive feedback from you, which has undoubtedly strengthened my manuscript. Please find our point-by-point responses below.

Sincerely,

∗Significant revisions have been made throughout the manuscript, including the correction of ambiguous expressions written in passive voice. Especially, the sections of the manuscript beyond part 4, which were not subject to your comments, have been revised as well.

*Significant revisions and enhancements have been made to the manuscript, including changes to the title.

∗Almost all the points raised have been addressed and incorporated into the revisions.

*The revised manuscript has been uploaded with the attachment.

Reviewer 2 Report

Comments and Suggestions for Authors

This article examines the process and specific context of erasing wonder and moving on to merit in state and individual rites, in the early Chosŏn Dynasty. This is different from existing studies that explained the reason for the reform of the rites as Confucianization.

In the early Chosŏn Dynasty, rites at both the state and individual levels were greatly innovated. At that time, rites at the state level were transformed into a legitimate and ethical act of repaying and thanking those who had done good deeds to the nation and its people. In addition, personal rites were understood as cultivating virtue and doing good deeds, and were limited to offering rites to one's ancestors. This was different from the existing religious perspectives that believed in a personal divine being with wonder and hoped to obtain what one wanted by relying on wonder.

It is explained that the reason for such heterogeneous changes different from the contemporary trend was due to the acceptance of the newly emerging religious perspectives in the Ming China and various measures based on it, although Confucian orientation may also have had an influence. However, this was far from the reality at the time, and it was not until the late Chosŏn Dynasty that changes in individual perspectives were achieved to some extent. However, state rites were not easily implemented in the early Chosŏn Dynasty, but the reform and exclusion of wonder of state rites came to an end with the reign of King Taejong and King Sejong, and this was seen to have served as the basis of state rites during the Chosŏn Dynasty.

However, even after King Sejong, there were many examples of the state or king using wonder, as in the case of King Sejo, which shows that wonder was still a powerful way to solve immediate problems. This means that existing practices have the property of being maintained over a long period of time rather than disappearing all at once. Therefore, evaluation of both aspects of continuity and change will be necessary.

Author Response

Dear a reviewer,

Thank you for the opportunity to revise my manuscript (Manuscript ID: religions-2851917) following your comments. I have thoroughly reviewed each comment and made careful revisions to my manuscript accordingly. Below, I provide a detailed response to each of your' suggestions and comments.

I appreciate the constructive feedback from you, which has undoubtedly strengthened my manuscript. Please find our point-by-point responses below.

Sincerely,

∗Comment: "evaluation of both aspects of continuity and change will be necessary."

∗Response and Revisions: The title, along with the overall content, has been revised, and a chapter has been added to comprehensively examine both aspects of continuity and change.

∗The revised manuscript has been uploaded with the attachment.

Reviewer 3 Report

Comments and Suggestions for Authors

-       Content is repetitive and paragraphs rather short. Provide more concrete examples that support your claims. 

-       Add a conclusion! (currently missing)

-       Make the content more accessible to readers outside of the Korean history field. For example, “people conducting rituals beyond their status” requires further explanation

-       Provide your definition of “wonder.” Include the Korean term/hancha. Could “miracle” be a better translation?

-       P. 3: wŏnguje: explain the purpose of this rite here (right now the explanation appears on p 7)

-       “Cho Pak and others” frequently appears in the text => could you briefly explain who these “others” are?

-       P. 6: secondly, …; thirdly, … => could you provide some examples for these two points? “various shrines” or “shrines in several places” is too vague. Please provide more concrete examples that spell out the names of the shrines and the context in which rituals took place at specific shrines.

-       P. 7: avoid one-sentence paragraphs. Either merge with other paragraph or delete.

-       P. 8 “Yi Chik and others” = delete “and others” or explain who they are.

-       P. 10 “individual level of rites” = what is the Korean term for it? meaning? How are these different from the state rite system?

-       P. 10: again, avoid one-sentence paragraphs

-       According to the new system, who was allowed to conduct rites to mountains and rivers? Only the king? The magistrates? Could you provide an example for an early Chosŏn yangban official or the king conducting rites to mountains and rivers?

Comments on the Quality of English Language

English could be improved. "wonder" bad word choice.

Author Response

Dear a reviewer,

Thank you for the opportunity to revise my manuscript (Manuscript ID: religions-2851917) following your comments. I have thoroughly reviewed each comment and made careful revisions to my manuscript accordingly. Below, I provide a detailed response to each of your' suggestions and comments.

I appreciate the constructive feedback from you, which has undoubtedly strengthened my manuscript. Please find our point-by-point responses below.

Sincerely,

*Significant revisions and enhancements have been made to the manuscript, including changes to the title.

∗All the points raised have been addressed and incorporated into the revisions.

*Especially, a section corresponding to the conclusion has been added, and the term "wonder" has been changed to "spiritual efficacy." The meaning of "spiritual efficacy" has been clarified using footnotes.

*Footnotes have been utilized to specify the meanings of 'state rites' and 'personal rites'.

*The revised manuscript has been uploaded with the attachment.

Round 2

Reviewer 1 Report

Comments and Suggestions for Authors

Second Revisions

  • Abstract

    • Reword:

      • "In the fifteenth century, the government of Chosŏn Korea,"

      • rites, eliminating...efficacy, aligning --> rites by eliminating...efficacy to align

      • despite their theoretical appeals --> despite being appealing theoretically

      • spiritual efficacy, achieved --> spiritual efficacy, which was achieved

  • pg 1

    • In Koryŏ... -> During the Koryŏ period

    • what sponsors wish--> what sponsors wished

    • were beings that had been granting --> were beings that granted

  • pg 2

    • and enforced that ---> and enforced rules where

    • Neo-Confucianism: the author needs a brief statement regarding what he means by Neo-Confucianism for the benefit of a non-Korean studies reader

    Here I will stop pointing out minor issues with English that could be fixed with some copyediting

    • understood in academia --> understood

    • w-->

  • pg 3

    • Ham... sajŏn

      • This is a blockquote and should be offset typographically

    • songhwang-> sŏnghwang

    • miraculously power--> miraculously powerful

  • pg 4

    • Koryŏ, from the sajŏn--> Koryŏ from the sajŏn

  • pg 7

    • beyond the Chosŏn king's status as a vassal-> beyond the Chosŏn king's prerogative as a vassal

    • conventional practices: Unlear in this context

  • Bibliography

    • Juhn Ahn's name is reversed; surname is Ahn.

    • Martina Deuchler 1992; surname is reversed

    • Bibliography has the order of western names incorrectly alphabetized:

      • Valerie Hansen; Martina Deuchler; Juhn Y. Ahn; Peter K. Bol;

    • Note that author is inconsistent in giving kanji names of Japanese authors.

Comments on the Quality of English Language

See above.

Author Response

"I sincerely appreciate the comments and suggestions provided from you,

I have incorporated all the comments and suggestions and revised the manuscript accordingly. The modified sections have been highlighted in red. In addition to this, I have also made some revisions to the manuscript on my own initiative.
